# Disparities in telemedicine use and payment policies in the United States between 2019 and 2023
Anna D. Gage [1], Megan A. Knight[1], Corinne Bintz[1], Robert W. Aldridge [1,2], Olivia Angelino [1], Joseph L. Dieleman[1,2], M. Ashworth Dirac[1,2,3], Laura Dwyer-Lindgren[1,2], Simon I. Hay[1,2], Rafael Lozano[1,2,4], Ali H. Mokdad[1,2] & Annie Haakenstad [1,2] ✉

## Abstract

**Background** The COVID-19 pandemic induced an increase in telemedicine use in the American health care system. We assess disparities in telemedicine usage, the diseases and conditions it is used for, and the association of payment parity policies with telemedicine use for January 2019–March 2023. **Methods** We include health systems which reported electronic health record data to the Healthjump database. The outcomes of interest are the percentage of outpatient consultations conducted via telemedicine in each health system and the distribution of outpatient and telemedicine consultations across 31 diseases and conditions. We use a difference-in-difference observational design to assess the association of state level payment parity mandates with telemedicine use. **Results** We show telemedicine use grew from less than 0.05% of outpatient consultations in 2019 to 25% in April 2020 and 4% in March 2023. Health systems in urban areas used telemedicine 2.4 times more than health systems in rural areas since April 2020 at the median. In March 2023, 29% of all mental health care visits and 21% of substance use disorder care were provided via telemedicine. Payment parity mandates are associated with a 2.5 percentage point increase in telemedicine use in the first quarter of 2023 compared to states without mandates. **Conclusions** The pandemic resulted in a sustained change in the use of telemedicine. The predominance of mental health care in telemedicine suggests that this mode of service delivery could be instrumental to increasing access to mental health services in the United States.

## Plain language summary

Telemedicine describes healthcare that is not provided in person but provided through electronic communications such as telephones or computers. The use of telemedicine increased during the COVID-19 pandemic. Our study aims to understand what kinds of health providers were providing telemedicine and for which health conditions from 2019 to 2023. We also examine whether policies for payment influenced telemedicine use. We found that providers in urban areas provided telemedicine more than those in rural areas, and telemedicine was most often used for mental health and substance use disorder care. Additionally, mandating clinicians be reimbursed at equivalent rates for telemedicine is associated with higher telemedicine use. Our findings indicate that telemedicine continues to have a role in American healthcare since the pandemic, which highlights the need for appropriate payment policies to be in place to ensure equitable provision.

The COVID-19 pandemic systematically disrupted health care in the United States. In April 2020, as providers and patients conformed with social distancing mandates, worked to decongest health care facilities, and minimized COVID-19 exposure, telemedicine use surged[1,2]. Telemedicine insurance claims grew 23-fold from January to June 2020, and by October 2020, 43% of Americans had used a telemedicine service in the prior six months[1,3]. Reports suggest that telemedicine continued to be used frequently into 2022[4–6].

An increase in telemedicine use is thus one potential lasting effect of the pandemic on the American health care system. Providers and patients are

now more accustomed to telemedicine as compared to prior to the pandemic. Protocols, technology, and norms are in place for its regular use in many settings[7,8]. With the conclusion of the official public health emergency on May 11, 2023[9], many telemedicine policies are still in flux and there are debates on how it should be used moving forward[10–12]. Key questions include which health systems use telemedicine, which diseases and conditions telemedicine is used for, and which policies incentivize telemedicine use.

First, a key question is which health systems provide care via telemedicine. Telemedicine can help make health care more convenient, reduce

[1]The Institute for Health Metrics and Evaluation, University of Washington, Seattle, WA, USA. [2]Department of Health Metrics Sciences, University of Washington, Seattle, WA, USA. [3]Department of Family Medicine, University of Washington, Seattle, WA, USA. [4]School of Medicine, National Autonomous University of Mexico, Mexico City, Mexico. ✉e-mail: ahaak@uw.edu

travel time, and minimize lost work hours, but primarily for those with digital literacy and good internet access—characteristics which are less common among older, low-income, racial and ethnic minority, and rural populations[13]. Research on telemedicine during the pandemic has shown lower frequency of telemedicine use among African American persons relative to non-Hispanic white persons, as well as lower use for rural residents and older adults[6,14,15]. However, while most studies focus on patient factors, there is a gap in understanding what kinds of health systems offer telemedicine, which may be the strongest predictor of use[16].

Second, it is important for clinicians to develop best practices around what kinds of care telemedicine should be used for. Some interactions cannot be conducted virtually (e.g., physical exams) but others, such as counseling, can be delivered via telemedicine with comparable effectiveness[17,18]. The existing literature on telemedicine is often fragmented by specialty; for example, studies have separately examined the trends of telemedicine use for mental health care[19], diabetes care[20], HIV care[21], and oncology[22,23]. However, we are unaware of any studies to date which compare the mode of care provision for all specialities throughout the duration of the pandemic. Having a systematic understanding of how telemedicine is currently being used can help clinicians across specialties start to develop principles for how and when to deploy this modality.

Finally, there are ongoing policy discussions at the state and federal levels around how much telemedicine should persist and how to pay for it[10,11,24]. Prior to the pandemic, unclear or lower reimbursement rates for telemedicine were cited as a barrier to adoption[25]. The regulatory environment for telemedicine shifted rapidly early in the pandemic to include payment parity policies, which refer to whether providers are reimbursed for telemedicine by the payer at the same rate as they would for a comparable in-person service. Medicare and most state Medicaid programs instituted payment parity at the beginning of the pandemic, as did six of seven major commercial insurers[26,27]. Some states also mandated payment parity for all commercial insurers, both prior to and during the pandemic. Stable, clear, and equivalent reimbursement rates for telemedicine may have incentivized providers to scale up telemedicine rapidly during the early stages of the pandemic and invest in technology and adaptation for its use longer term. Several studies have shown increased telemedicine use in states with payment parity policies relative to states without those policies for contraceptive visits, newly diagnosed cancer care, among community health centers and in patient surveys[28–30]. However, there is a gap in understanding how payment parity policies affected telemedicine provision in the health system more broadly and whether this increase differed by health system characteristics starting at the onset of the pandemic in March 2020.

In this study, we tackle these three questions about the ongoing role of telemedicine. Using a large electronic health records database, we characterize the health systems where telemedicine was provided and for what diseases and conditions from January 2019 until March 2023. We also compare telemedicine use, with a focus on the early pandemic period and thereafter, across states that did or did not implement payment parity laws. We find that telemedicine grew substantially during the COVID-19 pandemic; by March 2023 it remained more common than prior to the pandemic. Health systems in urban areas used telemedicine as a share of outpatient visits at a higher rate than health systems in rural areas during the study period and care for mental health and substance abuse was most common. Payment parity policies were associated with a small increase in telemedicine use in the first quarter of 2023. Overall, our study aims to provide policymakers and clinicians with a perspective on who is providing telemedicine for what and the role of payment parity policies in supporting telemedicine use.

## Methods
### Data and participants
We utilized electronic health record (EHR) data that were aggregated by Healthjump, a data management platform that standardizes inputs between different EHR vendors. The Healthjump database includes records from 1327 health systems and 117 million unique patients across 50 states, Puerto Rico, and Washington, DC, collectively referred to here as states[14]. These data are comparable in characteristics to the nationally representative 2019 National Ambulatory Medical Care Survey[31] (Supplementary Table 1) and have been used in several published peer-reviewed studies[14,32,33]. Pro-bono access to the Healthjump data were supplied by the COVID-19 Research Database partners. This study was deemed not human subjects research by the Institutional Review Board of the University of Washington (#11304), which waived the need for informed consent and approved the set up and use of the data by co-authors.

For this analysis, we included data on encounters from January 1, 2019, to March 31, 2023, the last month available from Healthjump. Our unit of analysis is a health system, captured by all patient visits recorded in a shared EHR platform, which we take to represent commonly owned or managed health facilities and/or provider groups. Each encounter in Healthjump was associated with a particular health system identifier. To limit the impact of new units joining or leaving the database, we included only systems that report at least 50 consultations per year during 2019–2022 and at least one consultation during the first three months of 2023. We further excluded health systems that did not report any data on patient race or ethnicity or had improbable race or ethnicity values. We additionally excluded inpatient and emergency room visits and patient records missing three-digit ZIP code, gender, or birth year information (< 0.1% of patients). Exclusions are depicted in Supplementary Fig. 1.

### Variable definitions
All CPT procedure codes were grouped into consultations, defined as all procedures that occurred for a single patient on a single day. The condition or disease causing care was identified using the primary diagnosis code from the International Classification of Diseases Tenth or Ninth Revision (ICD-10 or ICD-9). The diagnosis codes were grouped into 31 categories of conditions using methods previously described[34]. Telemedicine was identified by use of CPT modifier codes 93, 95, GT, FQ, or FR for any procedure in a consultation. These modifier codes refer to synchronous audio-visual or audio-only communication between a patient and provider[35]. Of the used telemedicine modifiers in the data, 99.5% were codes 95 or GT, which both refer to care provided by interactive audio and video communication systems. However, documented flaws in the data do not allow us to definitively distinguish between audio-only and audio-visual communication[36].

Patient race and ethnicity are reported in the EHR. These were grouped into five mutually exclusive groups: Hispanic (of any race); white (non-Hispanic); Black (non-Hispanic); Asian, Native Hawaiian or Other Pacific Islander (non-Hispanic) (AAPI); and American Indian or Alaska Native (non-Hispanic) (AI/AN), consistent with the Office of Management and Budget 1977 standards[37]. We imputed race and ethnicity using multiple imputation and ten imputations with a logistic regression model for ethnicity and polytomous logistic regression for race using the following covariates: patient age, gender, and primary language spoken (English, Spanish, or language of Asian origin); the total number of consultations during the study period; the health system; and demographic and economic characteristics of the patient's three-digit ZIP code, including percentage non-Hispanic white, non-Hispanic Black, Asian, Hispanic, under age 18, over age 65, poverty rate, and rural. The demographic and economic characteristics were estimated from the 2019 American Community Survey ZIP code tabulation areas, except for percentage rural, which was from the 2010 Census[14]. Urban area is defined as incorporated place or territory with at least 2500 residents, based on Census criteria[38]. We reviewed the imputation model using a validation procedure and undertook a complete case analysis as a robustness check (Supplementary Methods 1, Supplementary Table 2, Supplementary Fig. 2).

Because we were interested in how payment affects providers' decisions to offer telemedicine, payment parity mandates were determined by the health system's state. As the locations of health systems and their associated facilities were unavailable, we assigned each health system to the state where most of its associated patients resided. States with payment parity had statutes or emergency orders requiring telemedicine reimbursement for

commercial insurers at the "same rate" or "same basis" as that for in-person services[39–41]. A map and dataset of the payment parity policies by state are included in Supplementary Fig. 3 and Supplementary Data 1.

States that mandated payment parity may have also implemented other mandates and stay-at-home orders related to the pandemic that would affect telemedicine use. We defined two variables using data from the Institute for Health Metrics and Evaluation's COVID-19 modeling database: the number of days per month a state implemented stay-at-home orders, and a monthly mandate propensity index[42]. The index was constructed first by calculating the length of time that each state had each of the following policy mandates in place: closures of bars, restaurants, gyms, and schools; mask mandates; and gathering restrictions. These indicators were then summarized into a single index using the first component of a principal components analysis to reflect both the duration and comprehensiveness of a state's pandemic-related mandates.

We examined how the association between telemedicine payment parity policies and telemedicine use varied by select health system characteristics during the early pandemic period and thereafter. Systems were grouped according to whether they were above or below the median value across all health systems. The following median cut-points were used: rurality of 25%, 75% of patients white Non-Hispanic, patient load of 1300 patients per month, and a mandate propensity score of 0.155.

## Statistics and reproducibility

We first described telemedicine provision over the study period, estimating the percent of all outpatient encounters that were conducted via telemedicine by patient race and by health system characteristic. We then described the distribution of diseases and conditions for which care was sought by whether it was provided in person or via telemedicine over the study period. Noting the high use of telemedicine for mental disorders before and during the pandemic, we next explored the trends in these services by mode of service provision.

We used a two-way fixed effects difference-in-difference design to assess whether payment parity laws increased the use of telemedicine during the COVID-19 pandemic. Defining the outcome as the percentage of outpatient consultations per quarter conducted via telemedicine, we compared use in health systems in states that had mandated payment parity by April 2020 with those in states that never mandated payment parity. Using the model written in Supplementary Methods 2, we estimate the effect of parity over time (measured in quarters) after controlling for time-invariant system characteristics, quarters, and a series of time-varying system characteristics. These characteristics include the percentage of patients per quarter who were white, Black, Hispanic, under age 18, over age 65; the log number of patients per quarter; the percentage change in the patients per quarter from the 2019 average; the mandate propensity index; and stay-at-home orders. We further investigated the heterogeneity of the association of payment parity with telemedicine use by the select health system characteristics listed above using additional models with interaction terms. In the main analysis, we excluded systems in five states that implemented parity after April 2020, enabling us to avoid negative weights and other biases associated with heterogeneity in treatment timing[43]. All analyses were conducted on each of the ten datasets imputed for race and ethnicity and pooled using Rubin's Rules[44].

A set of robustness checks were conducted. First, we assigned systems in states that mandated payment parity after June 2021 to the control group and estimated the association for a more limited period, to July 2021, to understand how the exclusion of these systems in these states affects the results. Second, we summarized the data to the state level and estimated the association for states rather than health systems. Finally, we estimated the association among those who are under 65, given that the older population had payment parity mandated through Medicare.

## Reporting summary

Further information on research design is available in the Nature Portfolio Reporting Summary linked to this article.

## Results

Electronic health record data from 498 health systems are retained in the analysis (Table 1). The health systems are located across 43 states and Puerto Rico, in every region of the United States (Supplementary Fig. 4). Over the study period from January 2019 to March 2023, 5.0% of the 115.8 million outpatient consultations that these health systems provided were conducted via telemedicine. Systems in states that never mandated payment parity had fewer days of stay-at-home orders, were more rural, and had more white non-Hispanic patients than systems in states that mandated payment parity.

The percentage of outpatient consultations conducted via telemedicine grew from less than 0.05% during 2019 to a peak of 24.9% (95% CI: 24.8–25.0) during April 2020 (Fig. 1a). By March 2023, telemedicine made up 4.0% (95% CI: 3.8–4.1) of outpatient consultations, still 80 times higher than prior to April 2020. White patients had higher telemedicine use than Black or AI/AN patients during the April 2020 peak, but by January 2021, there were no differences in telemedicine use between these groups (Fig. 1b). Health systems in more urban areas provided telemedicine more often than systems in rural areas throughout the duration of the pandemic period; in the median month, urban health systems provided telemedicine care at 2.4 times the rate than rural health systems (Fig. 1c). Larger health systems provided telemedicine slightly more than smaller systems throughout the study period (Fig. 1d). States that had more pandemic-related mandates in place had higher telemedicine usage than states with fewer restrictions (5.1%, vs. 2.0% in March 2023, Fig. 1e).

The disease or condition used for telemedicine care is compared to the causes of in-person outpatient care in Fig. 2. Prior to the COVID-19 pandemic, when telemedicine was used very infrequently, mental health care, on average, made up 58.3% of telemedicine consultations. Beginning in April 2020, the distribution of care provided via telemedicine aligned more closely with the variety provided via in-person care as it became used more frequently, though with fewer well visits (13.5% in-person, vs. 2.6% via telemedicine) and more mental health care (4.8% in-person vs. 15.7% via telemedicine).

Mental health care was the single most common type of care provided via telemedicine; the increase in telemedicine use in April 2020 corresponds with an overall increase in delivery of mental health care and substance use disorder care, which increased by 2424 overall consultations in April 2020 compared to March 2020 (95% CI: 2230–2618) (Fig. 3). In March 2023, 28.7% (95% CI: 28.4–30.0) of mental health care and 21.2% (95% CI: 20.2–22.1) of substance use disorder consultations were provided via telemedicine, in contrast with 3.9% (95% CI: 3.9–3.9) of all other outpatient care.

The distribution of mandated payment parity across included health systems over time is shown in Supplementary Fig 5. In comparison with systems in states that never mandated payment parity, in Q2 2020 telemedicine use was 6.6 percentage points (95% CI: 4.9–8.2) higher in systems in states that mandated payment parity and 2.5 percentage points (95% CI: 0.9–4.1) in Q1 2023 after controlling for time-varying factors, quarter, and health system (Fig. 4 & Supplementary Data 2). Comparing systems with and without mandated parity, the increases in telemedicine use associated with mandates were larger in systems with more racial and ethnic minority patients, in areas that were more urban, and in smaller systems. Small systems in states that mandated parity had 14.5 percentage points (95% CI: 11.4–17.8) higher telemedicine use than small systems in states than never mandated parity in the Q2 2020; this difference dropped to 2.9 percentage points (95% CI: -0.3 to 6.1) by Q1 of 2021. Similarly, systems in states that mandated parity and had high mandate propensity had 9.5 percentage points (95% CI: 5.8–13.0) higher telemedicine usage in Q2 2020 than states that mandated parity but had fewer other mandates. Systems in more rural areas had lower telemedicine use in payment parity states than rural systems in never-parity states.

Results of the sensitivity checks are shown in Supplementary Figs. 6-8. These results were robust to treating states that mandated parity after June 2021 as never-parity states. The increases in telemedicine use in mandated

**Table 1 | Sample characteristics**

|  | Parity never mandated | Parity mandated by April 2020 | Parity mandated after April 2020 | Overall sample |
|---|---|---|---|---|
| **State characteristics** |  |  |  |  |
| Number of states | 21 | 18 | 5 | 44 |
| Mandate propensity index, mean (sd) | 0.14 (0.06) | 0.19 (0.07) | 0.14 (0.09) | 0.16(0.07) |
| Stay at home days, mean (sd) | 36 (24) | 51 (52) | 42 (23) | 43 (38) |
| **System characteristics** |  |  |  |  |
| Number of health systems | 324 | 137 | 37 | 498 |
| Average visits per month, median (IQR) | 1113 (2603) | 1516 (2918) | 1738 (3924) | 1293 (2991) |
| Median rurality of area (IQR) | 33% (32%) | 12% (36%) | 23% (31%) | 27% (34%) |
| Median poverty rate of area (IQR) | 15% (5%) | 14% (6%) | 15% (6%) | 15% (5%) |
| **Patient characteristics** |  |  |  |  |
| Number of patients | 9,450,942 | 4,191,278 | 959,575 | 14,601,795 |
| Age |  |  |  |  |
| Under 15 | 14% | 14% | 12% | 14% |
| 15-64 | 64% | 63% | 64% | 64% |
| 65 Plus | 22% | 22% | 23% | 22% |
| Female | 55% | 55% | 57% | 55% |
| Race and ethnicity |  |  |  |  |
| White non-Hispanic | 72% | 61% | 59% | 68% |
| Black non-Hispanic | 16% | 11% | 17% | 15% |
| Hispanic of any race | 9% | 23% | 18% | 14% |
| Asian, Native Hawaiian or Other Pacific Islanders | 2% | 5% | 5% | 3% |
| American Indian or Alaskan Native | <1% | <1% | 1% | <1% |
| Total consultations per patient 2019-2023, mean (SD) | 8.5 (12) | 7.5 (11) | 7.8 (11) | 8.2 (12) |
| **Consultation characteristics** |  |  |  |  |
| Number of consultations | 77,903,479 | 30,634,746 | 7,305,427 | 115,843,652 |
| Telehealth | 4.2% | 6.5% | 6.0% | 5% |

parity states were larger among the population under 65 than the overall population.

## Discussion

The increase in telemedicine use induced by the pandemic represents a substantial and lasting change in the way that health care is provided in the United States. While telemedicine use has declined from its pandemic peak, it remained more widely used in early 2023 than in 2019. Despite its promise for expanding health care access in rural regions, telemedicine was consistently used more in urban areas. Telemedicine was frequently used for mental health care and substance use disorder care. Starting in April 2020, telemedicine consultations for mental health corresponded with an overall increase in the provision of mental health care rather than a substitute for in-person care.

We found that mandating parity in reimbursement was associated with higher telemedicine use during the COVID-19 pandemic. With a substantial pre-pandemic baseline, data that capture all types of patients and care, and a period of study that extends into 2023, our study provides more generalizable evidence than existing studies that similarly find payment parity policies were associated with more telehealth visits[5,28–30]. Our study also contributes substantially to understanding disparities associated with parity and telemedicine use: we found that payment parity mandates were associated with larger increases for systems in urban areas among systems with a higher proportion of racial and ethnic minority patients. White and AAPI patients switched to telemedicine more rapidly than patients of other race/ethnicities at the beginning of the pandemic, suggesting initial gaps in provision of telemedicine by health systems who primarily serve Black, Hispanic, and AI/AN populations, as well as demand barriers such as insurance status,

broadband access and language ability[3,45]. The reimbursement parity regulations may have created a stable environment for health care practices to continue investing in telemedicine technologies and systems and led to more sustained telemedicine use, particularly for smaller systems that have fewer resources. Given that telemedicine use was higher in urban areas, these mandates may reinforce disparities in telemedicine use, however.

These findings have several important policy implications. First, telemedicine usage is likely to continue to persist at higher levels into the future than prior to the pandemic. This is especially true for mental health and substance use disorder care, where telemedicine was a key mechanism through which care was delivered. Our study is the first to show that mental health and substance use disorder care were the most commonly provided services via telemedicine through 2023 for a population comparable to users nationwide, extending existing studies similarly showing increases in mental health care utilization corresponded with a rise in telemedicine care during the pandemic[19]. The question going forward is how effective this modality of care is for these disorders and whether a growing reliance on telemedicine for mental health care may eventually disadvantage populations in need in rural areas.

A second important policy question is thus how telemedicine policies can avoid exacerbating inequities in healthcare access. We found that payment parity policies were most effective in increasing telemedicine use among urban populations and small health systems, leaving populations in rural areas behind. Prior cross-sectional studies similarly found lower telemedicine use in rural versus urban areas, but no difference in the willingness to use telemedicine[46]. As payment parity policies alone do not appear sufficient in increasing telemedicine access in rural areas, future research should investigate which policies, such as expanding broadband

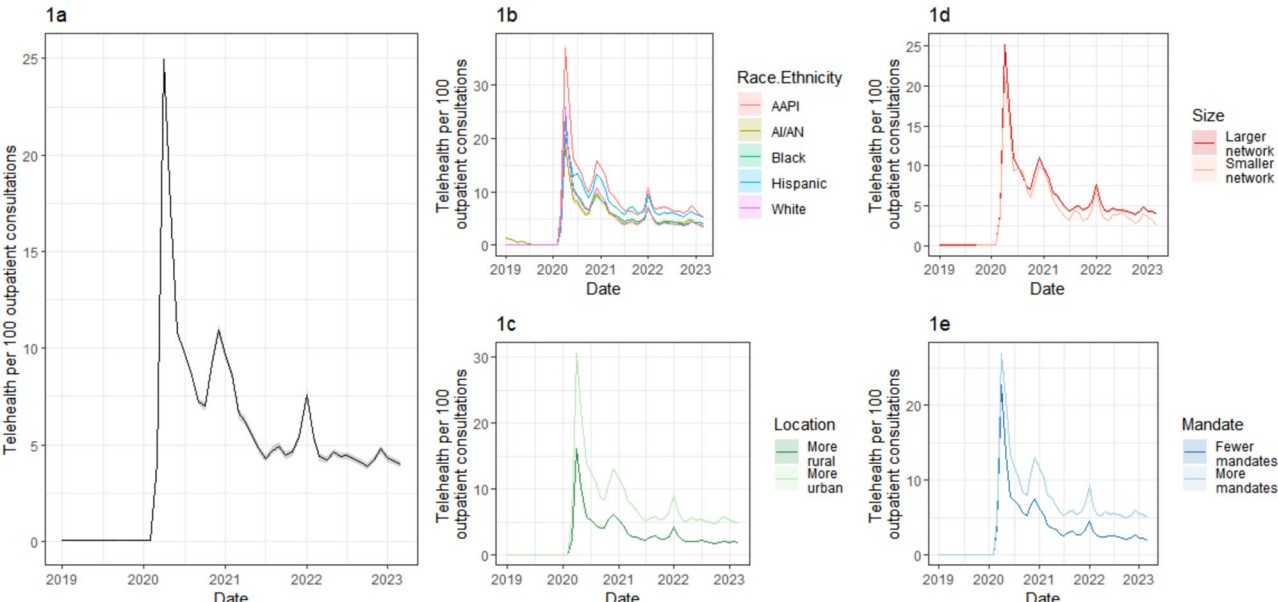

**Fig. 1 | Trends in telemedicine utilization from January 2019 to March 2023.**
**a** Overall trend per 100 outpatient consultations. **b** By race and ethnicity, showing telemedicine frequency among patients of each race/ethnicity group. AAPI: Asian, Native Hawaiian and Other Pacific Islanders (Non-Hispanic). AI/AN: American Indian and Alaska Native (Non-Hispanic). Black: Black non-Hispanic. Hispanic: Hispanic ethnicity of any race. White: White non-Hispanic. **c** By rurality of 3-digit ZIP code where most patients reside (split at median value of 25% rural), showing telemedicine frequency among health systems of each location group. **d** By size of the health system (split at median value of 1300 patients per month), showing telemedicine frequency among health systems of each group size. **e** By mandate propensity of the network's state (Split at median value of 0.155), showing telemedicine frequency among health systems of each group of states.

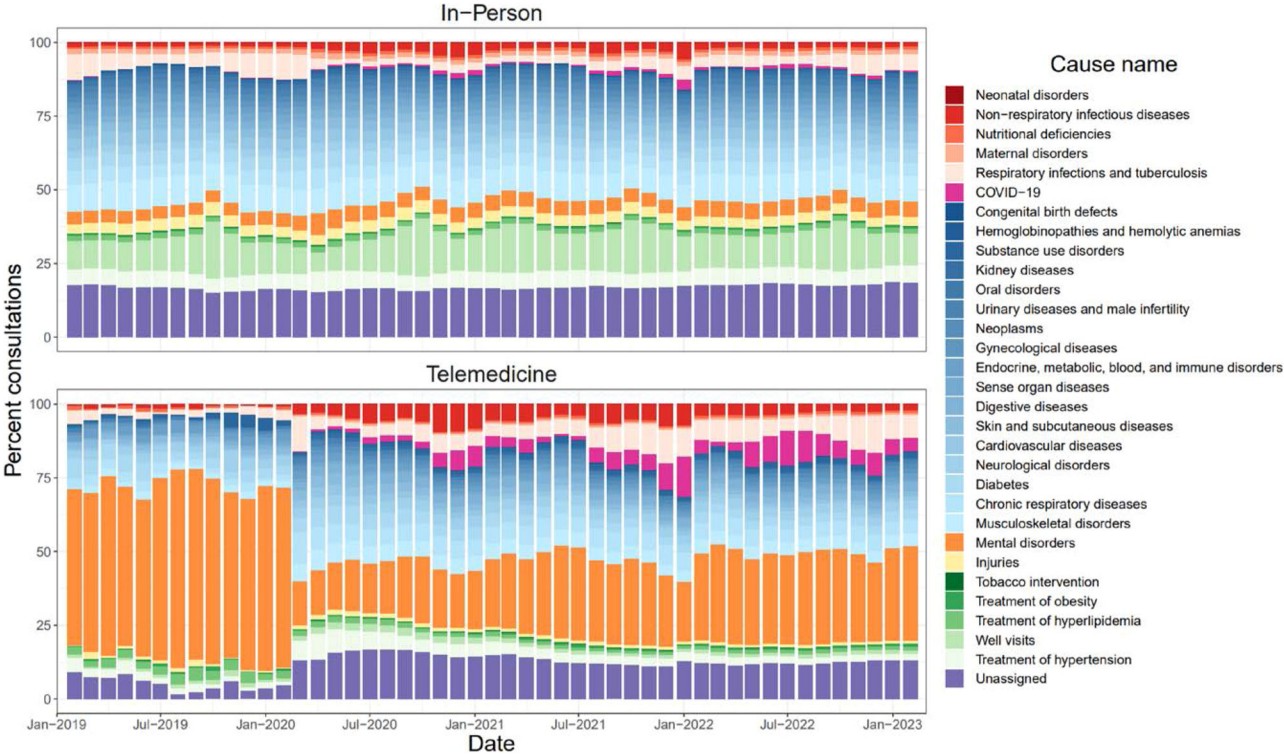

**Fig. 2 | Health conditions managed via telemedicine care and in-person outpatient care. a** Among all in-person consultations, the percent that were for care from 31 categories of conditions and diseases. **b** Among all telemedicine consultations, the percent that were for care from 31 categories of conditions and diseases.

networks or increasing digital literacy, could dismantle barriers to telemedicine in rural areas. A comparative perspective with telemedicine trends and disparities in other countries could offer valuable insights into best practices and innovative approaches in different health systems as they become available[11].

This study has several limitations. First, using the EHR data, we are not able to evaluate the quality of care provided via telemedicine versus in-person care. While telemedicine may expand access, there may also be declines in the quality of patient–provider interactions. This likely depends on the type of conditions—telemedicine may be well-suited for mental health services

**Fig. 3 | Modality of mental health care, substance use disorder, and all other care, January 2019 to March 2023. A** Number of encounters in thousands for mental health disorder care, separated by mode of delivery. **B** Number of encounters in thousands for substance use disorder care, separated by mode of delivery. **C** Number of encounters in thousands for all other care besides mental health and substance use disorder, separated by mode of delivery.

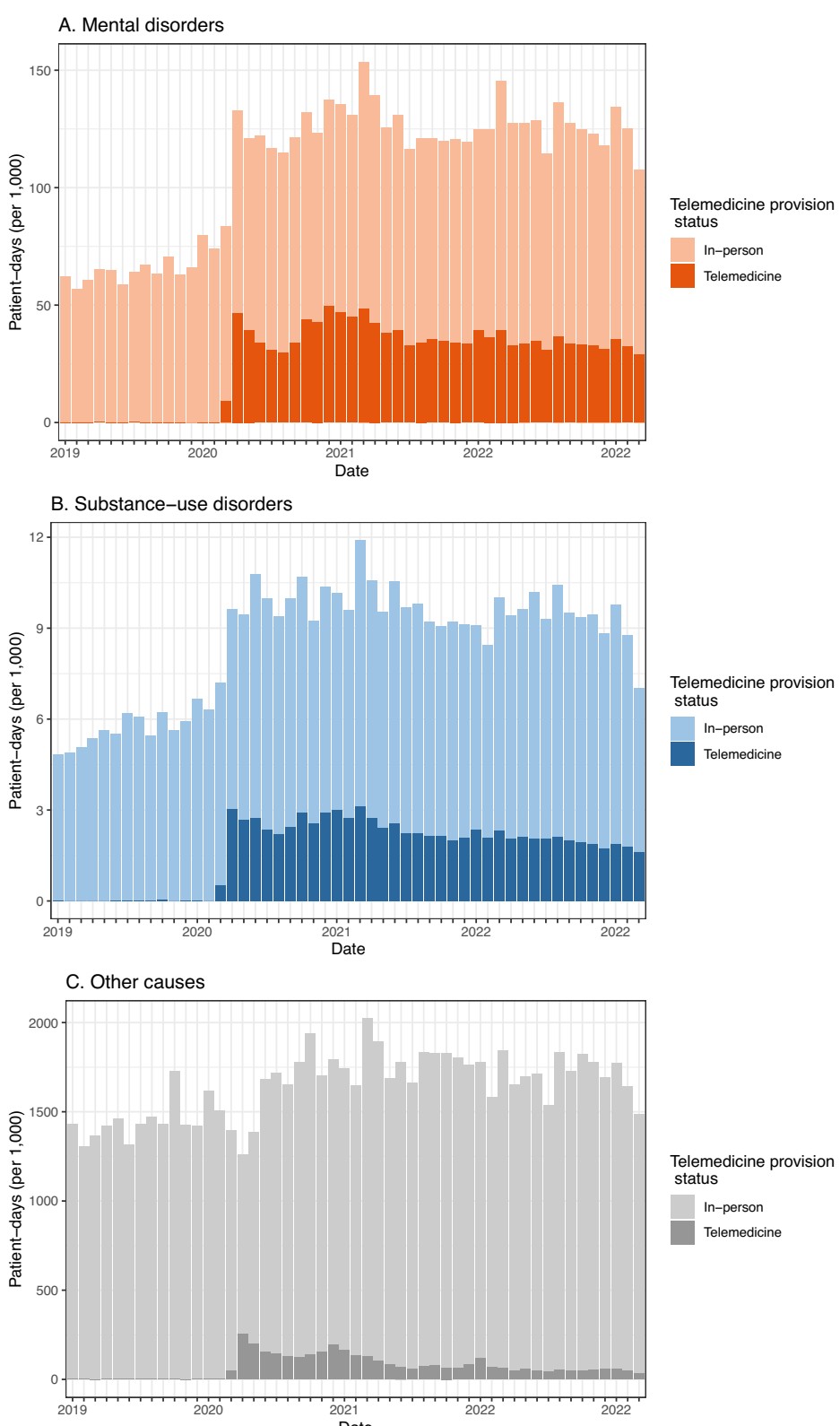

and chronic disease management, for instance[20,47]. Future research should investigate how patient outcomes differ with expanded use of telemedicine.

Second, because we are assuming the state a health system is located in based on patient location, there may be measurement error with respect to treatment assignment, in accidentally assigning some systems to states that do not reflect the correct payment parity assignment. This error would underestimate the association of parity on telemedicine usage; therefore, our findings are likely a conservative estimate. Third, we are unable to track patients across health systems or assess whether they sought additional health care outside of the included health systems. Fourth, our data, while large and covering most states in the U.S., is a convenience sample and not representative. Finally, while we implemented a rigorous design that controls for many potential confounders, unobserved covariates may still confound the association between parity polices and telemedicine provision.

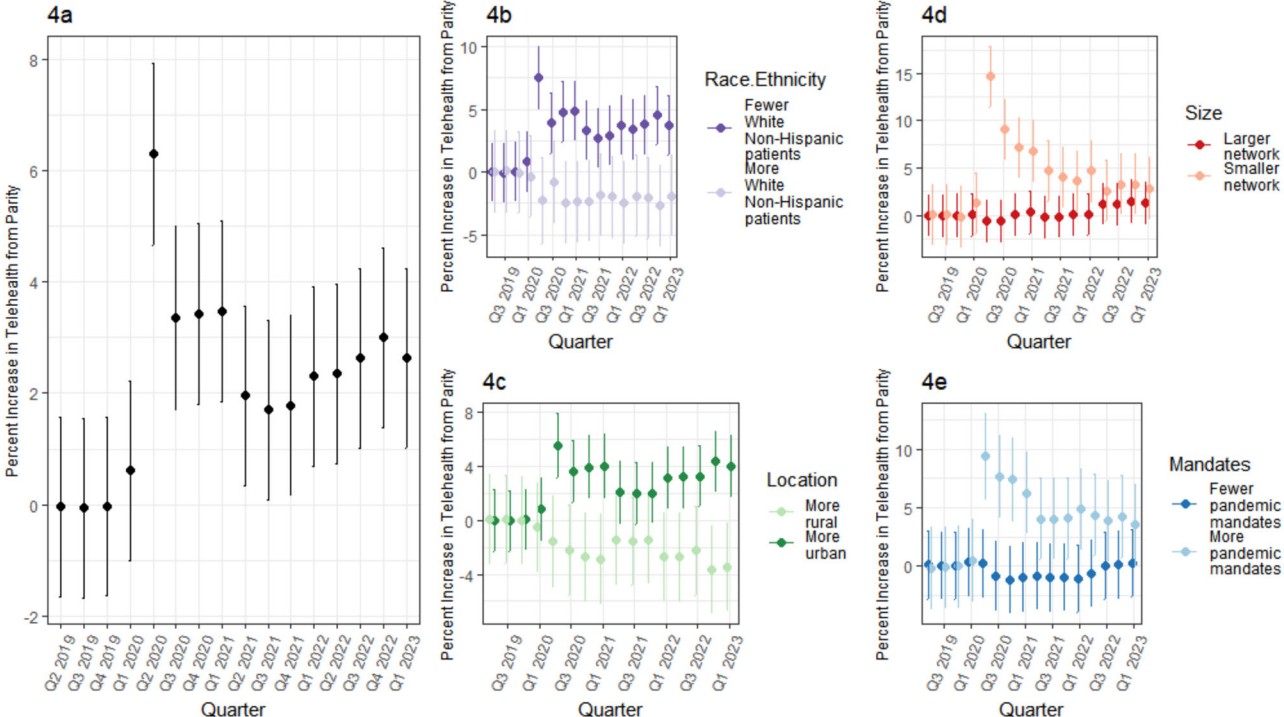

**Fig. 4 | Association of telemedicine payment parity mandates with telemedicine use by system characteristic. a** Overall. **b** By race and ethnicity of the patient population (split at the median value of 75% white non-Hispanic patients). **c** By rurality of 3-digit ZIP code where the modal patient resides (split at median value of 25% rural). **d** By size of the health system (split at median value of 1300 patients per month). **e** By mandate propensity of the network's state (Split at median value of 0.155). Error bars denote the upper and lower 95% confidence interval.

Our study underscores the post-pandemic persistence of heightened telemedicine use, with disparities across racial, ethnic, and geographical lines. We also discern a nuanced impact of payment parity policies, benefiting smaller, more urban, and more racially and ethnically diverse systems. Strategies beyond payment parity are needed to boost rural telemedicine usage as the US grapples with nationwide health system changes induced by the COVID-19 pandemic.

## Data availability
The secondary data used for this study are Electronic Health Record data including diagnosis, procedures and histories sourced from participating members of the Healthjump network. Data are not available publicly, but can be accessed via application to the COVID-19 Research Database[20,47]. Data underlying the main figures can be found in Supplementary Data 3. This analysis complies with the Guidelines for Accurate and Transparent Health Estimates Reporting (GATHER) statement, with further information provided in Supplementary Table 3[48].

## Code availability
Analyses were using R version 4.0.5. All code used for these analyses is publicly available online (https://github.com/ihmeuw/us_telemedicine).

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

## Acknowledgements

We received funding for this work from the Health Care Cost Institute, grant CORONAVIRUSHUB-D-21-00113. The data, technology, and services used in the generation of these research findings were generously supplied pro bono by the COVID-19 Research Database partners, who are acknowledged at https://covid19researchdatabase.org/. We thank Kelly Bienhoff, Adrienne Chew, Damian Santomauro and Kristofor Larsen for their contributions to this manuscript.

## Author contributions

Concept and design: Gage, Haakenstad Acquisition, analysis, or interpretation of data: Gage, Knight, Bintz, Angelino Drafting of the manuscript: Gage, Knight, Bintz, Haakenstad Critical review of the manuscript for important intellectual content: Gage, Knight, Bintz, Angelino, Aldridge, Dieleman, Dirac, Dwyer-Lindgren, Hay, Lozano, Mokdad, Haakenstad Statistical analysis: Gage, Knight Obtained funding: Haakenstad Administrative, technical, or material support: Gage, Knight, Bintz, Angelino, Haakenstad Supervision: Haakenstad

## Competing interests

The authors declare no competing interests.
