## [Transparent Peer Review file · Communications Medicine]

Disparities in telemedicine use and payment policies in the United States between 2019 and 2023

Corresponding Author: Dr Anna Gage

Version 0:

Reviewer comments:

Reviewer #1

(Remarks to the Author)

Thank you for the opportunity to review this manuscript. This research uses national electronic medical record data to examine telemedicine use across patient populations, health conditions, and policy environments in the year before and three years after the COVID-19 outbreak. The manuscript makes an important contribution by documenting the health services for which telemedicine is most common, health system characteristics associated with telemedicine use, and the impact of state-level reimbursement parity policy on telemedicine use. My comments are as follows:

1. Introduction/discussion: this research could be better situated within the vast literature on telemedicine that has proliferated since the COVID outbreak. What is already known, and what are the novel contributions of this study. For example, is this the first study to document mental health and substance use disorder treatment as the most common services provided via telemedicine?
2. Page 5, line 38: Since the focus is on health system characteristics, not individual clinicians, I recommend changing the language throughout to make this clear eg: "... we characterize the health systems where telemedicine was provided..."
3. Methods should go before results
4. Page 13, line 270: I'm curious about whether findings would differ if analyses were conducted at the patient-level. Probably outside the scope of this study, but since health systems can span across states, the health system analysis may underestimate findings re: reimbursement parity. It would also be helpful to know how many multi-state health systems there were, and the extent to which services were provided in both parity and non-parity states.
5. It would be helpful to have more details on the purpose of and methods used to generate figures 2 & 3. Figure 2 is a little more intuitive, but it is unclear what the y-axis represents in figs 3a-c, these must be counts, not percents? This could also probably be fixed by labeling these more clearly and including notes.
6. Page 6, line 121: please use the term 'substance use disorder' instead of 'substance abuse' throughout the manuscript.
7. Page 7, line 128: It would be helpful to have a table in the appendix with point estimates and CIs from all DiD analyses in figure 4.
8. Page 8, line 154: Looking at figure 3b, it seems that telemedicine SUD consultations resulted in an overall increase, but it is not clear whether there was a formal statistical test of this.
9. Page 9, paragraph 2: This might be a good place to discuss why more white patients received telemedicine in the early months of the pandemic. This may seem obvious but is important to discuss explicitly since examining disparities was a main focus of this analysis.
10. Page 10, line 202: "Regardless, we are unable to identify a causal relationship between payment parity and telemedicine use" This sentence is confusing since the DiD results suggest a positive relationship between payment parity and telemedicine. Is this meant to suggest that you cannot make causal inference from the design?
11. Page 21, line 442: Do Figures 1b-c represent patient or health systems-level analyses? If the former, this should be stated explicitly in the methods.
12. Appendix Page 2, table 1: Is race/ethnicity imputed for the Healthjump sample here?
13. Appendix 2, figure 1: would be helpful to have the percentages here. I'm also curious about how many encounters (or patients) were excluded through health system exclusions.

Reviewer #2

(Remarks to the Author)

I enjoyed reading this carefully written paper. This paper would be more impactful if it were simplified. I do not think there is a

need to include extensive descriptive information about the recipients of telemedicine and the clinical conditions for which it is used; this information is well known. It would be more impactful if this were a more straightforward policy evaluation of the parity mandates. Even that, however, is not real novel. There are at least five papers demonstrating the impact of the parity policy, although I agree that this may be the most comprehensive across settings and conditions.

I would suggest that you are clearer about the goal of this paper and whether you are interested in evaluating the causal effect of this policy – I think you are. Alternatively, you might frame it as evaluating the magnitude of the effect attributable to the policy. I'm not sure what the goal was.

You make a lot of the finding that there was differential impact across different patient groups – I do not think that you included interaction terms in the model so you probably should not over interpret this observed heterogeneity in treatment effect.

Specific comments:

Line 70 “Finally, there are ongoing policy discussions at the state and federal levels around how much telemedicine should persist and how to pay for it.” The reference you cite is 3 years old – this doesn't motivate the manuscript well.

Line 50 what is meant by provider networks in this sentence

line 54 is the word providing supposed to be. That paragraph is poorly constructed

line 60 this paragraph is under referenced; might include doi: 10.3122/jabfm.2022.03.210229.

Methods

Line 231 the methodology for assigning patient visits to the health facilities or the provider groups – is that the methodology recommended by Healthjump or was this something designing by you and what do we know of the validity

Line 234 what was the impact of excluding health systems without race or ethnicity data

Line 272 does MODAL state mean – and is the the state of residence of the patients

Line 281 the description of the propensity index requires another one or two sentence more explanation

Line 286 what is the causal question of interest; the word influenced isn't an effective word; were you more interested in the magnitude of effect?

Line 298 You assigned health systems based on information in the future which you recognize causes some misclassification; I recognize that you addressed this in the supplementary material but I wonder why this was chosen as the main analysis? Why didn't you just let the states change their classification at the time of the parity law (or perhaps after a brief lag)?

Figure 4 – I find this hard to interpret as policy analysis as there isn't really a time zero given that many of the states had parity laws in place throughout your observation time. I'm not sure what to suggest.

Results and Discussion

Line 105 – I don't understand this sentence

Line 131-133 this suggests that there were interaction terms in the model but I don't think there were

Line 162 there are other relevant studies to cite here
J Am Coll Emerg Physicians Open. 2021 Jan 14;2(1):e212359.

JNCI Cancer Spectr. 2023 Aug 31;7(5):pkad072. doi: 10.1093/jncics/pkad072. \10.1002/emp2.12359. eCollection 2021 Feb.

Line 202 I don't understand this sentence that you were unable to identify a causal relationship between payment parity and telemedicine use-- wasn't that the goal of this study

Line 209 What is the expected impact of your finding that the payment parity policies have differential impact? What should policy makers do with that information?

Version 1:

Reviewer comments:

Reviewer #3

(Remarks to the Author)

Thank you for the opportunity to review this manuscript. This is a novel paper examining telemedicine use in the US during the COVID-19 pandemic, disparities in telemedicine use, and the impacts payment parities had. This is important to discuss as the pandemic changed a lot regarding our healthcare system and telemedicine was one of those changes. The results from this study help us to better understand the impacts on healthcare systems and how to move forward to continue to improve our healthcare system.

I feel the author adequately addressed all the past reviewers' comments and I have no concerns for this paper.

Reviewer #1 (Remarks to the Author):

Thank you for the opportunity to review this manuscript. This research uses national electronic medical record data to examine telemedicine use across patient populations, health conditions, and policy environments in the year before and three years after the COVID-19 outbreak. The manuscript makes an important contribution by documenting the health services for which telemedicine is most common, health system characteristics associated with telemedicine use, and the impact of state-level reimbursement parity policy on telemedicine use. My comments are as follows:

We appreciate your thoughtful review.

1. Introduction/discussion: this research could be better situated within the vast literature on telemedicine that has proliferated since the COVID outbreak. What is already known, and what are the novel contributions of this study. For example, is this the first study to document mental health and substance use disorder treatment as the most common services provided via telemedicine?

Thank you for this suggestion. We have expanded the discussion of the existing literature and its gaps in the introduction to better position how our research fits within the literature. In the discussion, we have also tried to highlight more clearly the novel contributions of our work.

2. Page 5, line 38: Since the focus is on health system characteristics, not individual clinicians, I recommend changing the language throughout to make this clear eg: "... we characterize the health systems where telemedicine was provided..."

Thank you for this suggestion, we have amended the text accordingly.

3. Methods should go before results

Revised according to Communications Medicine format.

4. Page 13, line 270: I'm curious about whether findings would differ if analyses were conducted at the patient-level. Probably outside the scope of this study, but since health systems can span across states, the health system analysis may underestimate findings re: reimbursement parity. It would also be helpful to know how many multi-state health systems there were, and the extent to which services were provided in both parity and non-parity states.

Thank you for these suggestions. There have been other studies examining patient-level associations of telemedicine with payment parity, for example, Lee & Singh (2023) and Ellison et al (2022) that are cited in our work and align with our findings for an overall increase in parity states relative to non-parity states. However, given that the decision to offer telemedicine care is one that is often made at a health system level, we decided that a health-system level analysis is where we could add the most value.

We also agree that the misclassification of health systems spanning multiple states likely underestimate the findings on payment parity. However, the data is structured in Healthjump such that we don't have the location of a particular health facility or health system, just the states where the system's associated patients live. Where we see health systems with patients in multiple states, we cannot determine whether it is due to patients crossing states to seek care, patients moving across states during the study period (only one state per patient is recorded), or a health system having locations in multiple states. We note this as a limitation in the analysis, and acknowledge that our estimates on parity reimbursement are more conservative than if we were able to eliminate this source of misclassification. This aligns with Lee and Singh (2023)'s findings, which while using a much different dataset at the patient level, found larger increases associated with parity than we did.

5. It would be helpful to have more details on the purpose of and methods used to generate figures 2 & 3. Figure 2 is a little more intuitive, but it is unclear what the y-axis represents in figs 3a-c, these must be counts, not percents? This could also probably be fixed by labeling these more clearly and including notes.

Thank you for this suggestion. We have clarified the purpose of these figures in the methods section, added notes to them and corrected the label of the y-axis in Figure 3.

6. Page 6, line 121: please use the term 'substance use disorder' instead of 'substance abuse' throughout the manuscript.

Thank you for the suggestion, which we have changed throughout.

7. Page 7, line 128: It would be helpful to have a table in the appendix with point estimates and CIs from all DiD analyses in figure 4.

Excellent idea, we have included this as Supplementary Table 3.

8. Page 8, line 154: Looking at figure 3b, it seems that telemedicine SUD consultations resulted in an overall increase, but it is not clear whether there was a formal statistical test of this.

We have conducted a statistical test and found an overall increase of 2424 SUD consultations (95% CI: 2230 – 2618) between March and April 2020, this is now added to the text.

9. Page 9, paragraph 2: This might be a good place to discuss why more white patients received telemedicine in the early months of the pandemic. This may seem obvious but is important to discuss explicitly since examining disparities was a main focus of this analysis.

Thank you for the great suggestion. We have added a sentence to this effect in the discussion section:

“White and AAPI patients switched to telemedicine more rapidly than other race/ethnicities at the beginning of the pandemic, suggesting initial gaps in provision of telemedicine by health systems who primarily serve Black, Hispanic, and AI/AN populations, as well as demand barriers such as insurance status, broadband access and language ability.”

10. Page 10, line 202: “Regardless, we are unable to identify a causal relationship between payment parity and telemedicine use” This sentence is confusing since the DiD results suggest a positive relationship between payment parity and telemedicine. Is this meant to suggest that you cannot make causal inference from the design?

We have revised this statement to more precisely state our concern that there may still be unobserved confounding in the relationship between parity policies and telemedicine provision.

11. Page 21, line 442: Do Figures 1b-c represent patient or health systems-level analyses? If the former, this should be stated explicitly in the methods.

We have amended the notes of Figure 1 to clarify and described in the methods. Figure 1b is a patient level analysis as health systems have patients of many race/ethnicities. Figures 1c-1e are health systems level analyses. This is also reflected in the results text.

12. Appendix Page 2, table 1: Is race/ethnicity imputed for the Healthjump sample here?

Yes, we have added a note to indicate this and that the data prior to imputation is presented in Supplementary Methods 1.

13. Appendix 2, figure 1: would be helpful to have the percentages here. I’m also curious about how many encounters (or patients) were excluded through health system exclusions.

Thank you for these suggestions; we have amended the figure accordingly

Reviewer #2 (Remarks to the Author):

I enjoyed reading this carefully written paper. This paper would be more impactful if it were simplified. I do not think there is a need to include extensive descriptive information about the recipients of telemedicine and the clinical conditions for which it is used; this information is well known. It would be more impactful if this were a more straightforward policy evaluation of the parity mandates. Even that, however, is not real novel. There are at least five papers demonstrating the impact of the parity policy,

although I agree that this may be the most comprehensive across settings and conditions.

I would suggest that you are clearer about the goal of this paper and whether you are interested in evaluating the causal effect of this policy – I think you are. Alternatively, you might frame it as evaluating the magnitude of the effect attributable to the policy. I'm not sure what the goal was.

Thank you for this thoughtful feedback. We have expanded the discussion of the existing literature in the introduction to clarify where the gaps are and how our study fits into this literature. For instance, in reviewing existing peer-reviewed studies, we had found that while there is some indication of conditions and diseases telemedicine is used for, it is far from comprehensive, instead focusing on particular conditions or settings rather than providing a holistic and comprehensive sense of what conditions and diseases underpin use of telemedicine across types of care. We thus added the following text to explain our justification of that focus:

“The existing literature on telemedicine is often fragmented by specialty; for example, studies have separately examined the trends of telemedicine use for mental health care,¹⁹ diabetes care,²⁰ HIV care,²¹ and oncology.^{22,23} However, we are unaware of any studies to date which compare the mode of care provision for all specialities throughout the duration of the pandemic.”

Furthermore, we believe our paper makes an important contribution because it focuses on the characteristics of the health systems providing telemedicine, not individual patient characteristics. While most studies focus on patient characteristics, we found a dearth of studies examining which types of health systems are providing this care. Given that the decision to offer telemedicine originates from health systems and the supply side explains much of the variation in receipt of telemedicine according to prior studies, we think this description is an important contribution to the literature.

Finally, we agree that there is existing literature on the role of payment parity, and we have expanded our reference to these studies, thank you for the suggestion. However, we note that the existing literature tends to fall short of being comprehensive, instead focusing on particular sub-populations, not the patient population at large. We added text to that effect as follows:

“Several studies have shown increased telemedicine use in states with payment parity policies relative to states without those policies for contraceptive visits, newly diagnosed cancer care, among community health centers and in patient surveys.^{28–30} However, there is a gap in understanding how payment parity policies affected telemedicine provision in the health system more broadly and whether this increase differed by health system characteristics.”

We also agree with the reviewer that this paper has a wide scope but feel this is a key strength of our study. It is vital that description of the health systems and health

conditions used for telemedicine be considered alongside the role of payment parity to ensure policymakers and researchers have an understanding of who benefits from the existing policy landscape and who has further to gain by expanding payment parity or other policies supporting telemedicine. Tackling these research objectives in a comprehensive way using a single large dataset provides a much more comprehensive picture of telemedicine than has been available to date. For example, our study enables understanding of how payment parity could support increased mental health care access. This is a finding that would not be possible without the combination of these research objectives in one study. Another example is that the heterogeneity analysis in combination with the descriptive statistics on health system provision of telemedicine use highlights the key policy finding that rural health systems are being left behind in telemedicine provision and other policies beyond payment parity would be necessary to support telemedicine provision in these systems.

We very much appreciate the reviewer's perspective on simplifying this piece but believe our contribution is in the strength of the multiple analyses and providing them together to make sense of heterogeneity in the telemedicine landscape across the US.

You make a lot of the finding that there was differential impact across different patient groups – I do not think that you included interaction terms in the model so you probably should not over interpret this observed heterogeneity in treatment effect.

As you queried, we indeed included interaction terms in additional models to support the heterogeneity analysis. We have clarified the methods and supplementary materials to make this clear.

Specific comments:

Line 70 "Finally, there are ongoing policy discussions at the state and federal levels around how much telemedicine should persist and how to pay for it." The reference you cite is 3 years old – this doesn't motivate the manuscript well.

Good point, we have added additional and more recent citations to support this point.

Line 50 what is meant by provider networks in this sentence

Good point, we revised to use the 'health systems' language in line with the rest of the manuscript

line 54 is the word providing supposed to be. That paragraph is poorly constructed

Thank you for the suggestion. We revised this to more clearly state the question, as follows:

"First, a key question is which health systems provide care via telemedicine."

line 60 this paragraph is under referenced; might include doi:
10.3122/jabfm.2022.03.210229.

Thank you, we have added this citation.

Methods

Line 231 the methodology for assigning patient visits to the health facilities or the provider groups – is that the methodology recommended by Healthjump or was this something designing by you and what do we know of the validity

We have clarified in the text that the encounters were identified to a particular health system in the Healthjump database.

Line 234 what was the impact of excluding health systems without race or ethnicity data

135 health systems (10%) were excluded because they did not have any race or ethnicity, a small share of the health systems studied. The below table shows other patient characteristics before and after exclusion of these health systems. Excluded health systems had greater proportions of older patients.

	Before exclusion	After exclusion
Female	57%	55%
Age <15	10%	14%
Age 15-64	54%	64%
Age 65+	34%	22%

We conducted a payment parity mandate sensitivity analysis in which we restricted only to patients under 65 in order to assess the effect of these exclusions. The results in the graph below are largely consistent with the overall results, suggesting that this shift in age did not meaningful affect the results.

Association of parity mandates with telemedicine use in patients under 65

If health systems contained data on race or ethnicity that met our criteria, they were included in the analysis and any missing values were multiply imputed. Our cross-validation analysis found that our imputation procedure was 76% accurate for race and 89% accurate for ethnicity.

Line 272 does MODAL state mean – and is the the state of residence of the patients

We have reworded in the text that this means the state where most of its associated patients resided.

Line 281 the description of the propensity index requires another one or two sentence more explanation

We have added more detail to the methods section accordingly:

“The index was constructed first by calculating the length of time that each state had each of the following policy mandates in place: closures of bars, restaurants, gyms, and schools; mask mandates; and gathering restrictions. These indicators were then summarized into a single index using the first component of a principal components analysis to reflect both the duration and comprehensiveness of a state’s pandemic-related mandates.”

Line 286 what is the causal question of interest; the word influenced isn't an effective word; were you more interested in the magnitude of effect?

We have reworded this sentence as below and also noted the heterogeneity analysis later in the Statistical Analysis section to better clarify.

“We examined how the association between telemedicine payment parity policies and telemedicine use varied by select health system characteristics during the early pandemic period and thereafter.”

Line 298 You assigned health systems based on information in the future which you recognize causes some misclassification; I recognize that you addressed this in the supplementary material but I wonder why this was chosen as the main analysis? Why didn't you just let the states change their classification at the time of the parity law (or perhaps after a brief lag)?

We appreciate this thoughtful question that is also useful for clarification. We are not assigning information based on the future – it's that we consider the combination of factors as our exposure, some of which were in place in the period prior to the onset of the pandemic but, from our perspective, required the pandemic to be fully taken advantage of or activated by providers and patients. There was such minimal pre-COVID use of the telemedicine modality that it is the combination of the onset of the pandemic along with payment parity that is the force behind the changes in utilization that we are interested in. This approach also enabled us to avoid negative weights and other biases associated with heterogeneity in treatment timing noted in the two-way fixed effects literature. We clarify this with the following underlined addition (p. 5):

“We also compare telemedicine use, with a focus on the early pandemic period and thereafter, across states that did or did not implement payment parity laws.”

We also had noted this previously as our focus in the *Statistical Analysis* section (p. 9) as below:

“We used a two-way fixed effects difference-in-difference design to assess whether payment parity laws increased the use of telemedicine during the COVID-19 pandemic.”

We did explore alternatives to the two-way fixed effects model, for example in using inverse proportional weights to control for health system characteristics and allowing parity status to vary to estimate the average treatment effect, as in the graph pasted below. While these results are broadly robust to the estimates presented in the paper, we chose to use the two-way fixed effect model in order to better control for time invariant factors as well as the dramatically changing telemedicine use over the course of the study period.

Figure 4 – I find this hard to interpret as policy analysis as there isn't really a time zero given that many of the states had parity laws in place throughout your observation time. I'm not sure what to suggest.

Several states did have payment parity policies in place before January 2019, so that they were 'treatment' states during the entire study period, while others remain 'controls' throughout the study period. We find that prior to the pandemic, parity policies did not meaningfully increase telemedicine use, but that policies did have an impact once the pandemic started and throughout the rest of Q1 2023. We take this to mean that onset of the pandemic meaningfully changed the policy landscape around telemedicine. While there is no time-zero for all states, we believe we were able to effectively capture the effect of parity by quarter as the pandemic progressed.

Results and Discussion

Line 105 – I don't understand this sentence

We have reworded to clarify:

“Health systems in more urban areas provided telemedicine more often than systems in rural areas throughout the duration of the pandemic period; in the median month, urban health systems provided telemedicine care at 2.4 times the rate than rural health systems (Fig 1c).”

Line 131-133 this suggests that there were interaction terms in the model but I don't think there were

We apologize for the confusion; we did in fact include interaction terms in the subsequent models. We have added this to the methods and Supplementary Methods 2 to clarify.

Line 162 there are other relevant studies to cite here
J Am Coll Emerg Physicians Open. 2021 Jan 14;2(1):e212359.

JNCI Cancer Spectr. 2023 Aug 31;7(5):pkad072. doi: 10.1093/jncics/pkad072.
\10.1002/emp2.12359. eCollection 2021 Feb.

Thank you for sharing these references. Given our focus on the post-pandemic period, we have included the second citation.

Line 202 I don't understand this sentence that you were unable to identify a causal relationship between payment parity and telemedicine use-- wasn't that the goal of this study

We have revised this statement to more precisely state our concern that there may still be unobserved confounding in the relationship between parity policies and telemedicine provision:

“Finally, while we implemented a rigorous design that controls for many potential confounders, unobserved covariates may still confound the association between parity polices and telemedicine provision.”

Line 209 What is the expected impact of your finding that the payment parity policies have differential impact? What should policy makers do with that information?

Thank you for these important questions. We have expanded the paragraph on policy implications of the differential impact of payment parity policies to better underline this point:

“A second important policy question is thus how telemedicine policies can avoid exacerbating inequities in healthcare access. We found that payment parity policies were most effective in increasing telemedicine use among urban populations and small health systems, leaving populations in rural areas behind. Prior cross-sectional studies similarly found lower telemedicine use in rural versus urban areas, but no difference in the willingness to use telemedicine. As payment parity policies alone do not appear sufficient in increasing telemedicine access in rural areas, future research should investigate which policies, such as expanding broadband networks or increasing digital literacy, could dismantle barriers to telemedicine in rural areas. A comparative perspective with telemedicine trends and disparities in other countries could offer valuable insights into best practices and innovative approaches in different health systems as they become available.”